# Hepatoprotective Efficacy of Cycloastragenol Alleviated the Progression of Liver Fibrosis in Carbon-Tetrachloride-Treated Mice

**DOI:** 10.3390/biomedicines11010231

**Published:** 2023-01-16

**Authors:** Theerut Luangmonkong, Pittaya Puphancharoensuk, Varisara Tongsongsang, Peter Olinga, Warisara Parichatikanond

**Affiliations:** 1Department of Pharmacology, Faculty of Pharmacy, Mahidol University, Bangkok 10400, Thailand; 2Centre of Biopharmaceutical Science for Healthy Ageing (BSHA), Faculty of Pharmacy, Mahidol University, Bangkok 10400, Thailand; 3Department of Pharmaceutical Technology and Biopharmacy, University of Groningen, 9713 AV Groningen, The Netherlands

**Keywords:** cycloastragenol, CCl_4_, liver fibrosis, hepatoprotection, fibrosis resolution

## Abstract

The continuous death of hepatocytes induced by various etiologies leads to an aberrant tissue healing process and promotes the progression of liver fibrosis and ultimately chronic liver diseases. To date, effective treatments to delay this harmful process remain an unmet clinical need. Cycloastragenol is an active phytochemical substance isolated from *Astragalus membranaceus*, a plant used in traditional Chinese medicine to protect the liver. Therefore, our study aimed to elucidate the efficacy of cycloastragenol on carbon-tetrachloride (CCl_4_)-induced liver fibrosis in mice. We found that cycloastragenol at 200 mg/kg dosage exhibited anti-fibrotic efficacy as demonstrated by a decrease in collagen deposition, downregulation of mRNA expression of collagen type 1, and a reduction in the content of total collagens. In addition, cycloastragenol further augmented the levels of anti-fibrotic matrix metalloproteinases (Mmps), that is, Mmp8, proMmp9, and Mmp12, which play a pivotal role in fibrosis resolution. According to histological analysis and serum markers of hepatotoxicity, cycloastragenol protected the livers from damage and mitigated the increment of serum alanine aminotransferase and bilirubin implicating hepatoprotective efficacy against CCl_4_. Moreover, cycloastragenol upregulated the mRNA expression of interleukin 6, a pleiotropic cytokine plays a vital role in the promotion of hepatocyte regeneration. In conclusion, cycloastragenol alleviated the progression of liver fibrosis in CCl_4_-treated mice and its anti-fibrotic efficacy was mainly due to the hepatoprotective efficacy.

## 1. Introduction

Repeated hepatic cell deaths resulting from chronic inflammation and oxidative stress lead to an aberrant tissue healing process which could promote liver fibrosis and ultimately chronic liver diseases. Persistent liver injury can be induced by various etiologies such as viruses [1], alcohol [2], drugs [3], cholestasis [4], and steatosis [5]. Following the necrosis and apoptosis of hepatocytes, these harmful contributors commence a series of events including the activation of quiescent hepatic stellate cells (HSCs) into fibrogenic myofibroblasts which induce excessive extracellular matrix (ECM) accumulation, especially fibrillar collagen type 1 [6,7]. In fact, ECM can be properly degraded leading to fibrosis resolution by enzyme matrix metalloproteinases (Mmps); however, an imbalance between the process of deposition and degradation may alter the composition of ECM proteins which eventually lead to scar formation and dysfunction of the affected liver tissue [8]. Besides necrosis and apoptosis, chronic inflammation and oxidative stress may trigger compensatory proliferation of mature hepatocytes and telomere shortening. As a result, hepatocytes are senescent, and liver regeneration is defective [9]. In the pathogenesis of liver fibrosis, these processes are regulated by several profibrogenic cytokines; however, the transforming growth factor-beta 1 (TGF-β1) signaling plays a pivotal role since it is responsible for the activation of myofibroblasts and the regulation of ECM homeostasis [10].

Cycloastragenol is an active phytochemical substance in *Astragalus membranaceus* (Fisch.) Bunge, huang qi, a plant used in traditional Chinese medicine to improve immune functions and protect the liver [11]. Several attractive pharmacological properties of cycloastragenol which might contribute to the beneficial effects on the liver including hepatoprotective efficacy, antioxidative and anti-inflammatory properties, and telomerase activation to elongate telomere, have been demonstrated [11]. Recently, a previous study in rats revealed that astragaloside, which is the parent compound of cycloastragenol prior to the hydrolysis process, could prevent bile duct ligation-induced liver fibrosis via the modulation of notch signaling [12]. In addition, a similar study in rats demonstrated that a combination of total astragalus saponins and glycyrrhizic acid alleviated both bile duct ligation and dimethylnitrosamine-induced liver fibrosis via the modulation of TGF-β1 pathways [13]. Nevertheless, the potency of cycloastragenol which is believed to be the biological active component of astragaloside on the amelioration of liver fibrosis induced by hepatotoxins are currently unknown. Therefore, our study aimed to elucidate the efficacy of cycloastragenol on carbon-tetrachloride (CCl_4_)-induced liver fibrosis in mice. Beyond anti-fibrotic potency, the effects of cycloastragenol on hepatoprotection, inflammation, oxidative stress, telomere length, and TGF-β1-related signaling were elucidated to explore the associated mechanism of action.

## 2. Materials and Methods

### 2.1. Animals

Male ICR outbred mice at 6 weeks old were purchased from the National Laboratory Animal Center, Nakhon Pathom, Thailand. The mice were housed in a temperature and humidity-controlled room with a 12 h light/dark cycle. The standard rodent diet and filtered water were supplied ad libitum. The study commenced after an acclimatization period of the mice in the housing room for at least 7 days. This study, which complied with the ARRIVE (Animal Research: Reporting of In Vivo Experiments) guidelines [14], was approved by the Animal Ethical Committee of the Faculty of Pharmacy, Mahidol University (PYR002/2021 and PYR008/2022).

### 2.2. Experimental Protocol

The mice were randomly divided into 4 groups (n = 10 per group, 40 mice in total): (1) normal, (2) control, (3) cycloastragenol 50 mg/kg, and (4) cycloastragenol 200 mg/kg. To induce liver fibrosis, mice in groups 2–4 were administered CCl_4_ (Shanghai Seasonsgreen Chemical, Shanghai, China) by intraperitoneal injection twice a week for a consecutive 8 weeks. The amount of CCl_4_ was gradually escalated from 0.03 mL/kg, 0.075 mL/kg, and 0.1 mL/kg in the 1st, 2nd, and 3rd weeks, respectively. During the 4th to 8th weeks, CCl_4_ was administered at 0.12 mL/kg. Prior to the administration, CCl_4_ was diluted in olive oil to inject with an equivalent volume according to the individual body weight of each mouse. Starting from the 5th week until the 8th week, cycloastragenol (King-tiger Pharm-Chem, Chendu, China), prepared by dispersing in 0.5% sodium carboxymethylcellulose, was given to the mice in groups 3 and 4 by oral gavage once a day for 5 days per week. The mice in group 2 were administered olive oil and 0.5% sodium carboxymethylcellulose in an equivalent amount was administered to groups 3 and 4. At 72 h after receiving the last dosage of CCl_4_, the mice were anesthetized using carbon dioxide before terminal cardiac puncture for blood collection until complete euthanization. Before the isolation, the livers were thoroughly perfused using 0.9% sodium chloride solution via the portal vein until there was no residual blood. The isolated liver was separated into several pieces for fixing in 10% neutral-buffered formalin and snap-freezing in liquid nitrogen for further assays.

### 2.3. Histological Evaluations

The median lobes of livers, preliminarily fixed in 10% neutral-buffered formalin for at least 48 h, were processed in a series of ethanol and xylene and embedded in paraffin. The 4 μm liver sections were stained with picro-sirius red to determine collagen fiber deposition [15]. In addition, another section of the same piece of livers was stained with hematoxylin/eosin for routine histopathological examination to evaluate the histological damage score, assessed from the degree of hepatocyte degenerations, necrobiotic changes, and infiltrated lymphocytes by a described method [16]. The sections were examined using an electric light microscope (Olympus IX-81, Tokyo, Japan). To analyze microscopical pictures, 5 random non-overlapping frames per liver were selected for the analysis using ImageJ (National Institutes of Health, Bethesda, MD, USA).

### 2.4. Serum Biomarker Measurements

The clotted blood was centrifuged at 10,000× g for 10 min to collect serum. The levels of liver-related injury markers, that is, alanine aminotransferase, aspartate aminotransferase, alkaline phosphatase, total and direct bilirubin, total protein, and albumin, were immediately quantified using an automated serum biochemical analyzer (Olympus AU400 Chemistry Analyzer, Tokyo, Japan) with the supplied diagnosis reagent kits.

### 2.5. Hepatic Hydroxyproline Assay

To quantify total hepatic collagens, a colorimetric assay to detect hydroxyproline, which is a unique modified-amino acid that is mostly found in collagens, was performed by a minor modification of the described procedure [17]. In brief, an exact weight of the snap-frozen liver (approximately 100 mg) taken from 2 different lobes was homogenized in 6 N hydrochloric acid and hydrolyzed at 95 °C for 16 h. The hydrolyzed samples were centrifuged and the supernatants were mixed with chloramine-T solution in citrate-acetate buffer pH 6.0 and isopropanol before being incubated in Ehrlich’s reagent (*p*-dimethylamino-benzaldehyde, perchloric acid, and isopropanol) at 60 °C for 1 h. The colorimetric product of the reaction was measured with a Synergy HT spectrophotometer (Agilent, Santa Clara, CA, USA) at 550 nm. The content of hydroxyproline in each sample was reported as μg hydroxyproline per 100 mg of liver or per the whole liver.

### 2.6. Total RNA and Genomic DNA Isolation

Total RNA and genomic DNA were isolated from a piece of snap-frozen livers (approximately 30 mg) acquired from the same lobe using the AllPrep DNA/RNA/Protein MiniKit (Qiagen, Venlo, The Netherlands). The quantification and qualification of RNA and DNA were measured using a Nanodrop One spectrophotometer (Thermo Fisher Scientific, Waltham, MA, USA). The isolated total RNA was consequently reverse-transcripted prior to a quantitative real-time polymerase chain reaction (PCR) and an RNA profiler for the evaluation of gene expression. The isolated genomic DNA was used to quantify the telomere length.

### 2.7. Evaluation of Gene Expression

Reverse transcription was performed using an RT^2^ First Strand Kit (Qiagen). In brief, the isolated RNA was initially incubated with a genomic DNA-eliminating buffer at 42 °C for 5 min. The DNA-eliminated RNA was reverse-transcripted at 42 °C for 15 min before stopping the reaction at 95 °C for 5 min.

Gene expression was determined using specific primers (Table 1) and a Brilliant III Ultra-Fast SYBR Green QRT-PCR Master Mix (Agilent) on a CFX96 Real-Time PCR Detection System (Biorad, Irvine, CA, USA) with a cycle at 50 °C for 10 min and at 95 °C for 3 min followed by 40 cycles of 95 °C for 15 s and 60 °C for 30 s. Expression levels were corrected using glyceraldehyde-3-phosphate dehydrogenase (GAPDH) as a reference gene (dCt) and compared with the control group (ddCt). The results are displayed as a fold induction (2^−ddCt^).

RT² Profiler™ PCR Array Mouse Fibrosis (Qiagen, GeneGlobe ID: PAMM-120Z) with preloaded primers in 100-well strips were used to elucidate 84 genes related to fibrosis. cDNA was mixed with a 2X RT^2^ SYBR Green ROX FAST Mastermix (Qiagen). Thermal cycling and fluorescence detection were performed on a Rotor-Gene Q (Qiagen) with a cycle of 95 °C for 10 min followed by 40 cycles of 95 °C for 15 s and 60 °C for 30 s. Expression levels were corrected using GAPDH as a reference gene. The results are displayed as the magnitude of gene expression when compared with normal using the company’s program for generating a heat-map clustogram.

### 2.8. Quantification of Telomere Length

Telomere length was measured using a Relative Mouse Telomere Length Quantification qPCR Assay Kit (Sciencell Research Laboratory, Carlsbad, CA, USA). In brief, the genomic DNA was mixed with a 2X GoldNStart TaqGreen qPCR master mix containing a primer set designed to recognize and amplify a specific part of telomere sequences. The single copy reference, designed to recognize and amplify a region on chromosome 10, was used for data normalization. The data are shown as relative telomere length when compared with the normal length using 2^−ddCt^ method.

### 2.9. Evaluation of Protein Expression

The expression of multiple Mmps and proteins associated with TGF-β1-related signaling was quantified using Milliplex immunoassay (Merck, Rahway, NJ, USA), a bead-based multiplex enzyme-linked immunosorbent assay (ELISA) which could analyze multiple target proteins simultaneously. In brief, a piece of snap-frozen livers (approximately 30 mg) acquired from the same lobe was homogenized in kit-supplied lysis buffer containing a phosphatase inhibitor with supplement of Protease Inhibitor Cocktail Set III (Calbiochem, San Diego, CA, USA). After the centrifugation of liver homogenate, the amount of target proteins in the supernatant was quantified using a specific conjugated antibody with a designed magnetic bead technology. The value of proteins was normalized using total protein content measured by a Pierce™ BCA Protein Assay Kit (Thermo Fisher Scientific).

### 2.10. Statistic

Data are expressed as means + standard error of the mean (SEM) of numerical results among the same treatment. The statistical tests on the means of different groups were performed using one-way analysis of variance (ANOVA) followed by Tukey’s multiple comparisons. Prism 6.01 (GraphPad Software, San Diego, CA, USA) was the software used for the statistical calculation. A *p*-value less than 0.05 was considered significant. For gene expression and relative telomere length, the statistical differences were determined on ddCt. Since some mice died during the induction of fibrosis, the number of animals per group at the end of experiment was 8–10.

## 3. Results

### 3.1. Anti-Fibrotic Efficacy

To evaluate the anti-fibrotic potency of cycloastragenol, fibrillar collagens by picro-sirius red staining, expression of collagen type 1 mRNA (*Col1a1*) by quantitative real-time PCR, and content of total collagens by hydroxyproline assay were performed (Figure 1). We found that fibrillar collagens were increasingly deposited in the livers of mice treated with CCl_4_. Similarly, at the gene level and the total content of collagens in the livers, CCl_4_ upregulated the expression of *Col1a1* and increased the hydroxyproline content, respectively. Based on all collagen-associated analyses, cycloastragenol at 200 mg/kg dosage exhibited anti-fibrotic efficacy in the liver of CCl_4_-treated mice as demonstrated by a decrease in collagen deposition, downregulation of *Col1a1* expression, and a reduction in the content of total collagens. In contrast, the anti-fibrotic efficacy of cycloastragenol at 50 mg/kg was not apparently observed, especially on the picro-sirius red staining and *Col1a1* expression. Therefore, the 200 mg/kg of cycloastragenol appeared to be the effective dosage, and the efficacy of this dosage was shown in the other analyses of our study.

Next, we screened the effects of cycloastragenol against CCl_4_ on the expression of 84 genes associated with fibrosis by using the PCR array and found that cycloastragenol may probably affect various pathways in the alleviation of liver fibrogenesis (Figure 2). Highlighted examples were the decreased expressions of collagen type 1-alfa 2 chain (*Col1a2*), collagen type 3 (*Col3a1*), and collagen maturation enzyme lysyl oxidase (*Lox*). Multiple subtypes of integrins (*Itga3*, *Itgb5*, *Itga1*, *Itgav*, *Itgb6*, *Itgb8*, and *Itgb1*) and integrin-linked kinase (*Ilk*) which are responsible for cell–cell and cell–ECM interactions were downregulated. In addition, cycloastragenol reduced the expression of the gene-encoding tissue inhibitor of metalloproteinases (*Timp1*, *Timp2*, and *Timp4*) while it increased the expression of Mmps (*Mmp3* and *Mmp9*). Beyond ECM-related genes, cycloastragenol also downregulated the expression of pro-inflammatory markers, such as chemokine ligands (*Ccl3*, *Ccl12*, and *Ccl11*), chemokine receptor 2 (*Ccr2*), interleukins (*Il1a* and *Il1b*), and signaling proteins (*Stat1* and *Nfkb1*). Vice versa, anti-inflammatory cytokines (*Il4*, *Il10*, and *Il13*) were upregulated. Furthermore, the expressions of markers related with TGF-β1 signaling-related markers (*Tgfbr2*, *Smad2*, *Tgfb1*, *Tgfb2*, *Ltbp1*, and *Tgif1*) and cell death (*Bcl2*, *Myc*, *Jun*, and *Akt1*) were decreased. Nonetheless, some markers (such as *Tnf*, *Il5*, *Pdgfa*, *Pdgfb*, and *Smad4*) were unexpectedly increased. Thus, other techniques were subsequently performed to elucidate the associated mechanism of cycloastragenol on fibrogenesis. 

Since the screening of genes relating to ECM remodeling was affected by cycloastragenol, the multiplex ELISA was used to assess the level of Mmps in the liver (Figure 3). We found that CCl_4_ obviously increased the expression of Mmp2, Mmp8, proMmp9, and Mmp12. Superior to the levels in the control group, cycloastragenol further augmented the levels of Mmp8, proMmp9, and Mmp12, indicating the elevation of fibrosis resolution.

### 3.2. Hepatoprotective Efficacy

The hepatocyte degenerations, necrobiotic changes, and infiltrated lymphocytes of the livers were assessed using hematoxylin/eosin staining (Figure 4). The histological analysis revealed that CCl_4_ induced a high degree of liver damage as seen by the spread of ballooning hepatocytes with condensed chromatin in the enlarged nucleus and infiltrated lymphocytes in the central area of the hepatic lobules. Significantly, cycloastragenol alleviated the harmful effects of CCl_4_; however, liver damage remained visible at a lower degree when compared with the control.

In line with the histological analysis, CCl_4_ evidently induced liver damages resulting in an increment in liver enzymes, especially alanine aminotransferase, in the serum. Moreover, the mice treated with CCl_4_ appeared to incite an impairment of liver function as seen by the increment of bilirubin, a waste biological product to be excreted by the liver (Figure 5). Cycloastragenol tended to mitigate the increment of serum liver enzymes, alanine aminotransferase, aspartate aminotransferase, alkaline phosphatase, and bilirubin, both measured as total (conjugated and unconjugated) and direct (conjugated). Thus, cycloastragenol elicited hepatoprotective efficacy and preserved liver function against CCl_4_.

### 3.3. Anti-Inflammatory, Antioxidative, and Anti-Senescent Efficacy

To investigate the mechanisms underlying hepatoprotective efficacy of cycloastragenol, several markers associated with inflammation, oxidative stress, and senescence were quantified. Among inflammatory-related markers, the mRNA expression of anti-inflammatory cytokine interleukin 6 (*Il6*) in the liver of mice who received CCl_4_ alone was significantly downregulated (Figure 6). In the mice who received cycloastragenol, the expression of *Il6* was recovered to be at the same level as expressed in normal mice. We could not detect an obvious alteration on other inflammatory-related markers, that is, insulin-like growth factor (*Igf*), peroxisome proliferator-activated receptor alpha (*Pparα*), and peroxisome proliferator-activated receptor gamma (*Pparγ*).

Focusing on antioxidative efficacy, we found that the gene expression of NAD(P)H quinone dehydrogenase 1 (*Nqo1*) was the most apparently upregulated in response to CCl_4_ (Figure 7). However, this cytoprotective enzyme was not remarkably affected by cycloastragenol. Besides the statistically insignificant alteration of the mRNA expression of nuclear factor erythroid 2-related factor 2 (*Nrf2*), the expression of genes encoding other enzymes responsible for scavenging reactive species was not significantly regulated by either CCl_4_ or cycloastragenol.

The gene expression of a senescent marker Bcl2-associated agonist of cell death (*Bad*) was not obviously increased in response to CCl_4_ (Figure 8). Thus, although the level was comparable to the normal, we could not conclude that cycloastragenol exhibited anti-senescent efficacy. This finding was in line with the length of telomere in the genomic DNA of mice treated with CCl_4_ and cycloastragenol.

### 3.4. Effects on TGF-β1 Signaling Pathway

Since the screening of gene expression revealed that several TGF-β1 signaling-related markers were affected by cycloastragenol, quantitative real-time PCR of TGF-β1 ligand (*Tgf-β1*) was conducted and the multiplex ELISA was used to assess the level of TGF-β1 signaling-related proteins in the liver (Figure 9). We found that CCl_4_ tended to increase the expression of *Tgf-β1* and phosphorylated protein kinase B (pAkt); however, the increments were not significantly higher than the normal or mice who received cycloastragenol. The expression of other TGF-β1 signaling-related markers was not remarkably changed by either CCl_4_ or cycloastragenol.

## 4. Discussion

To date, the sole therapeutic option for patients with advanced chronic liver diseases is liver transplantation. Nonetheless, this invasive and high-risk surgical procedure is sufficient for a limited number of patients [18]. Therefore, effective treatments remain an unmet clinical need. Among available options, the eradication of underlying causes which prevail the death of hepatocytes by using antiviral therapy for chronic hepatitis B and C infection could be considered the most effective treatment because the drugs can prevent or even reverse the progression of disease [19]. In traditional Chinese medicine, *Astragalus membranaceus* (Fisch.) Bunge has been used in various preparations to protect the liver from harmful causes for several centuries [20]. Therefore, cycloastragenol, which is the major active compound in this plant, is of high interest to target liver diseases [21]. The evidence from our study revealed for the first time that cycloastragenol alleviated the progression of liver fibrosis resulting from chronic exposure to a hepatotoxin CCl_4_. Moreover, we revealed that the hepatoprotective efficacy of cycloastragenol played a major role in the alleviation of fibrosis progression. Trichloromethyl free radical (CCl_3_·), which is the toxic metabolite of CCl_4_, induces liver damage by altered cellular integrity leading to swelling, cytolysis, and death of hepatocytes, and prolonged exposure of CCl_4_ establish liver fibrosis [22,23]. Following the death of hepatocytes, several pro-inflammatory and pro-fibrotic mediators such as chemokine ligands/receptors and TGF-β1, respectively, are released to aggravate the initial damage in contiguous hepatocytes, Kupffer cells, and HSCs [24]. Despite unclear in-depth mechanisms, we found that cycloastragenol mitigated the toxicity of CCl_4_, resulting in the reduction of leaked cytoplasmic liver enzymes in the serum. Furthermore, the hepatoprotective efficacy may contribute to the preserved metabolic function of the livers, since we found that the level of bilirubin which requires a phase II microsomal enzyme glucuronosyltransferase to be excreted [25] was reduced by cycloastragenol.

The resolution of excessive deposition of ECM is a pivotal function of Mmps, the master class of enzymes possessing protease activity that play a role in liver fibrogenesis [8]. Despite conventional dividends based on enzyme–substrate specificity and cellular locations, these proteases may be alternatively differentiated by their pathophysiological role into pro- and anti-fibrotic Mmps [26]. In our study, the level of a pro-fibrotic Mmp2 (gelatinase-A) [27] in the liver of mice who received cycloastragenol was slightly lower than mice treated with CCl_4_ alone. In contrast, the levels of anti-fibrotic Mmp8 (collagenase-2), Mmp9 (gelatinase-B), and Mmp12 (metalloelastase) were markedly increased by cycloastragenol. Our findings were in line with several animal studies targeting fibrosis [28,29,30].

Among the quantification of multiple markers associated with inflammation, oxidative stress, and senescence, we found that CCl_4_ significantly altered the mRNA expression of *Il6*. Even though Il6 is usually recognized as a deleterious mediator, recent evidence demonstrated that this pleiotropic anti-inflammatory cytokine plays a vital role in the promotion of liver regeneration in liver pathologies [31]. Moreover, a previous study in mice treated with CCl_4_ showed that a combination of Il6 and mesenchymal stem cell transplantation attenuated liver fibrosis in mice [32]. Thus, cycloastragenol, which upregulated *Il6* in our study, could possibly promote hepatocyte regeneration after the damage of CCl_4_. According to the effects on other markers using real-time quantitative PCR, anti-inflammatory activity would not be the main contributor to the hepatoprotective efficacy of cycloastragenol. Regarding markers of oxidative stress, the mRNA expression of *Nqo1* was increased by CCl_4_. A previous study showed that this cytoprotective enzyme played a role in the detoxification of reactive species in livers obtained from patients with paracetamol overdosage and primary biliary cholangitis [33]. Due to the fact that cycloastragenol did not significantly alter the expression of gene-encoding Nqo1, the antioxidant activity might be trivial for the beneficial efficacy of cycloastragenol in our study. Similarly, the anti-senescent activity of cycloastragenol may be negligible also. Nevertheless, cycloastragenol may possibly exhibit anti-fibrotic potency via anti-senescent activity in a case in which the experiment was performed in aged species [34].

Since we could not detect a significant alteration in the expression of transforming growth factor-beta receptor II, phosphorylated-Smad2 (pSmad2,), pSmad3, and Smad4 resulting from CCl_4_ exposure, the involvement of cycloastragenol on canonical TGF-β1 signaling could not be concluded. Although the canonical TGF-β1 signaling is usually recognized as the major activated pathway in liver fibrogenesis, a variety of responses in certain mouse strains against CCl_4_ were reported [17,35]. In our ICR mice, it is possible that cycloastragenol regulated non-canonical pathways of TGF-β1 signaling, due to the levels of phosphorylated-Akt being slightly affected by CCl_4_ and cycloastragenol. In addition, a previous study showed that an herbal extract ameliorated CCl_4_-induced liver fibrosis in mice by inhibiting Akt-mediated hepatocyte apoptosis and regulating farnesoid X receptor (FXR) activity [36]. The effects of cycloastragenol on FXR could be gleaned from another study targeting hepatic steatosis in diet-induced obesity mice. This animal study administered cycloastragenol as a diet supplement at a dosage of 100 mg/100 g diet. Since the C57BL/6 mice at 30 g body weight consumed a 2.5–3 g diet per day, these mice could receive cycloastragenol at approximately 80–100 mg/kg dosage [37]. The results from these obesity mice showed that cycloastragenol improved fatty liver via FXR activation. Unfortunately, this study did not assess outcomes relating to fibrosis. Nevertheless, the effective anti-fibrotic dosage of cycloastragenol at 200 mg/kg in our study could sufficiently activate FXR. Since several previous studies demonstrated that FXR agonists impeded liver fibrosis and inhibited hepatocyte apoptosis [38,39,40], FXR activation might partly contribute to the anti-fibrotic efficacy of cycloastragenol. This dosage correlation might imply how the low dosage at 50 mg/kg of cycloastragenol, which was selected from a study targeting skin inflammation in psoriatic mice [41], could not be sufficient for modulating FXR and provided a clear anti-fibrotic efficacy in our study. Furthermore, the notch signaling that was modulated by astragaloside in the prevention of liver fibrosis in bile duct-ligated rats [12] might also connect with the anti-fibrotic efficacy of cycloastragenol. 

Finally, it is worthwhile to mention that our study was performed on outbred mice. This fact could be considered as a limitation since a variation on the effects of CCl_4_ and cycloastragenol might be relatively large as we demonstrated in an acute toxicity model of CCl_4_ using these outbred mice [42]. On the other hand, although inbred mice are usually preferred in almost all biomedical research currently because of their reduced genetic variability, experiments in outbred mice may be considered a better choice since they consist of inter-individual genetic variation [43].

## 5. Conclusions

Cycloastragenol at the dosage of 200 mg/kg alleviated the progression of liver fibrosis in CCl_4_-treated mice. The anti-fibrotic efficacy of cycloastragenol was mainly due to its hepatoprotection and was partly derived by the increased ECM resolution resulting from the upregulation of anti-fibrotic Mmps. Although the major mechanism of action required further elucidation, the inhibition of the non-canonical TGF-β1/Akt signaling pathway and possibly the modulation of FXR were supposed to play a role contributing to the anti-fibrotic and hepatoprotective potency of cycloastragenol.

## Figures and Tables

**Figure 1 biomedicines-11-00231-f001:**
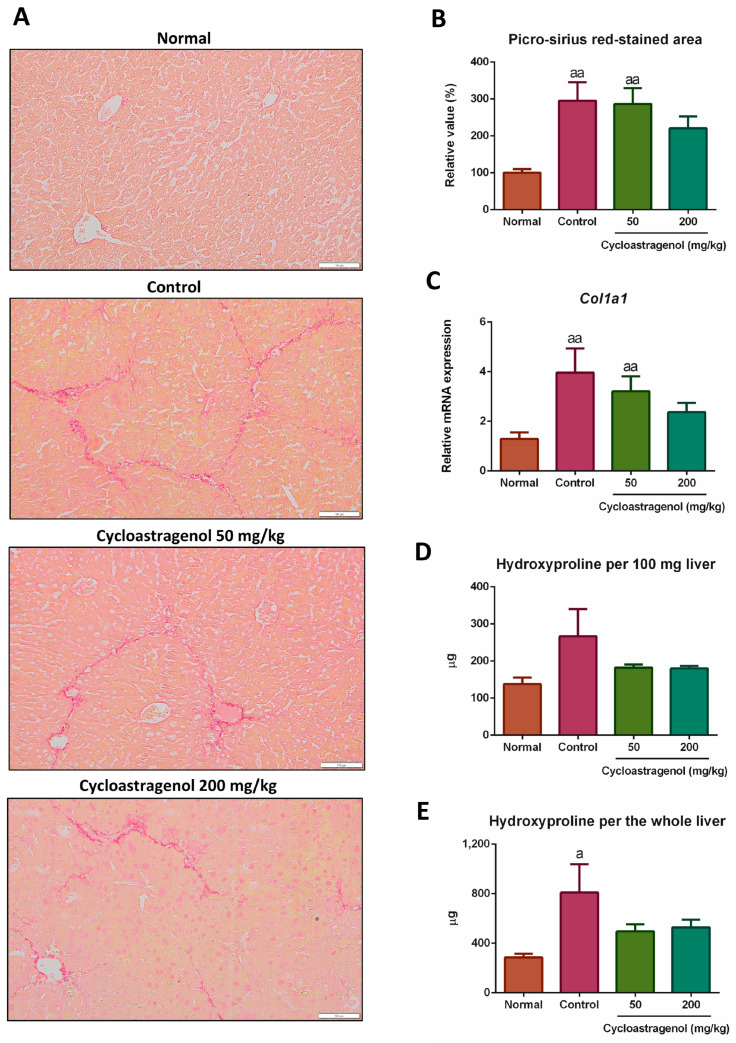
Anti-fibrotic efficacy of cycloastragenol (50 mg/kg and 200 mg/kg) in the liver of mice treated with carbon-tetrachloride. Representative picro-sirius red-stained pictures of the liver among each group (**A**) are shown. The stained areas of picro-sirius red (**B**) and relative mRNA expression of collagen type 1 (*Col1a1*, (**C**)) when compared with the normal are shown. Hydroxyproline contents of the liver when calculated per 100 mg liver tissue (**D**) and per the whole liver (**E**) are shown. Scaled bar = 100 μm. Bar graphs and corresponding error bars indicate means and SEM among the same treatment, respectively *(n* = 8–10). ^a^ and ^aa^ indicate *p* < 0.05 and 0.01 when compared with the normal, respectively.

**Figure 2 biomedicines-11-00231-f002:**
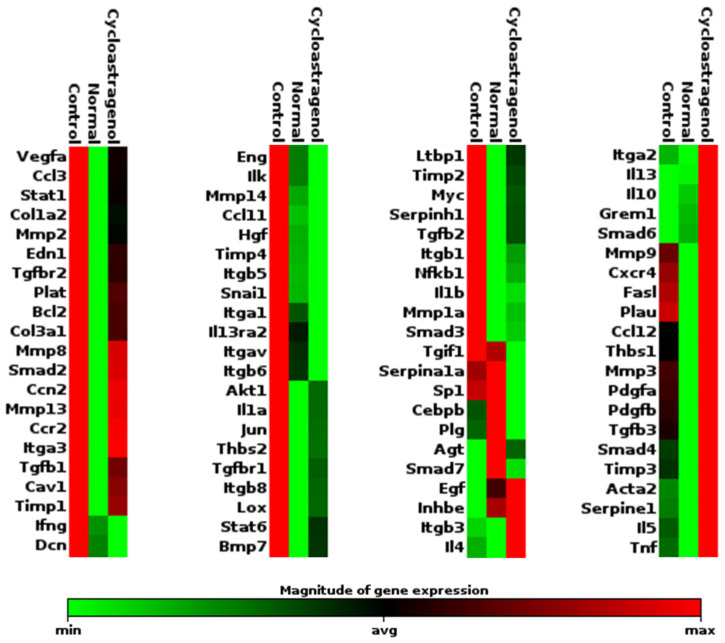
Effects of cycloastragenol (200 mg/kg) on the expression 84 fibrosis-related genes in the liver of mice treated with carbon-tetrachloride. Relative mRNA expression of each gene when compared with the normal are indicated using arrays of colors in a heat-map clustogram, representing a sample among each group, is shown.

**Figure 3 biomedicines-11-00231-f003:**
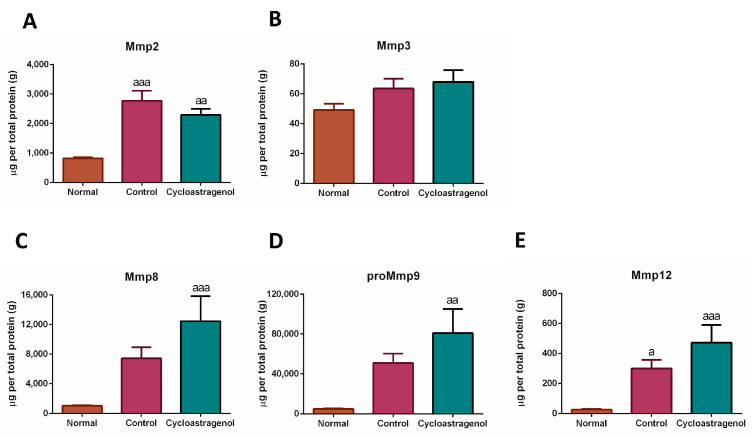
Effects of cycloastragenol (200 mg/kg) on the level of matrix-metalloproteinases (Mmps) in the liver of mice treated with carbon-tetrachloride. Data are expressed as the levels of Mmp2 (**A**), Mmp3 (**B**), Mmp8 (**C**), proMmp9 (**D**), and Mmp12 (**E**), per total protein content of the liver (g). Bar graphs and corresponding error bars indicate means and SEM among the same treatment, respectively (*n* = 8–10). ^a^, ^aa^, and ^aaa^ indicate *p* < 0.05, 0.01, and 0.001 when compared with the normal, respectively.

**Figure 4 biomedicines-11-00231-f004:**
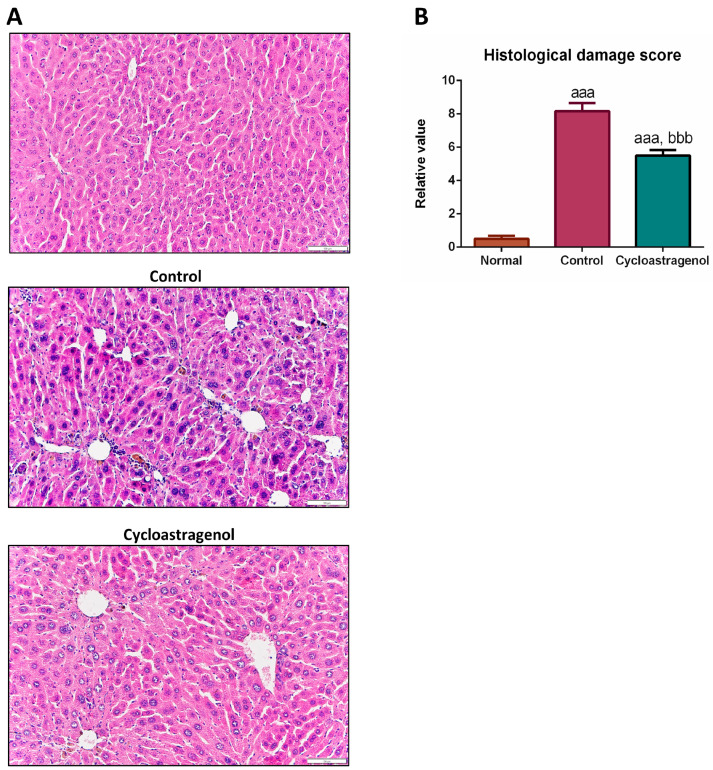
Hepatoprotective efficacy of cycloastragenol on histological damages in the liver of mice treated with carbon-tetrachloride. Representative hematoxylin/eosin-stained pictures of the liver among each group (**A**). The degree of hepatocyte degenerations, necrobiotic changes, and infiltrated lymphocytes were semi-quantified as histological damage scores. Data are shown when compared with the normal (**B**). Scaled bar = 100 μm. Bar graphs and corresponding error bars indicate means and SEM among the same treatment, respectively (*n* = 8–10). ^aaa^ and ^bbb^ indicate *p* < 0.001 when compared with the normal and control, respectively.

**Figure 5 biomedicines-11-00231-f005:**
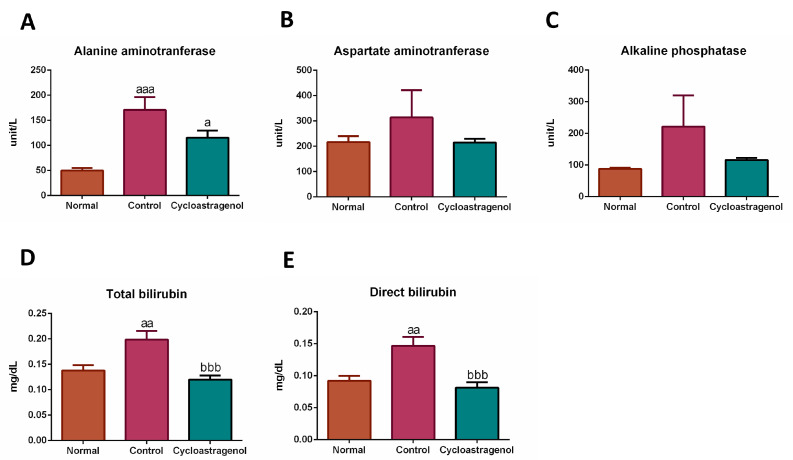
Hepatoprotective efficacy of cycloastragenol (200 mg/kg) on the levels of markers associated with liver toxicity in the serum of mice treated with carbon-tetrachloride. Levels of alanine aminotransferase (**A**), aspartate aminotransferase (**B**), alkaline phosphatase (**C**), total bilirubin (**D**), direct bilirubin (**E**), total protein (**F**), and albumin (**G**) are shown. Bar graphs and corresponding error bars indicate means and SEM among the same treatment, respectively (*n* = 8–10). ^a^, ^aa^, and ^aaa^ indicate *p* < 0.05, 0.01, and 0.001 when compared with the normal, respectively. ^bbb^ indicates *p* < 0.001 when compared with the control.

**Figure 6 biomedicines-11-00231-f006:**
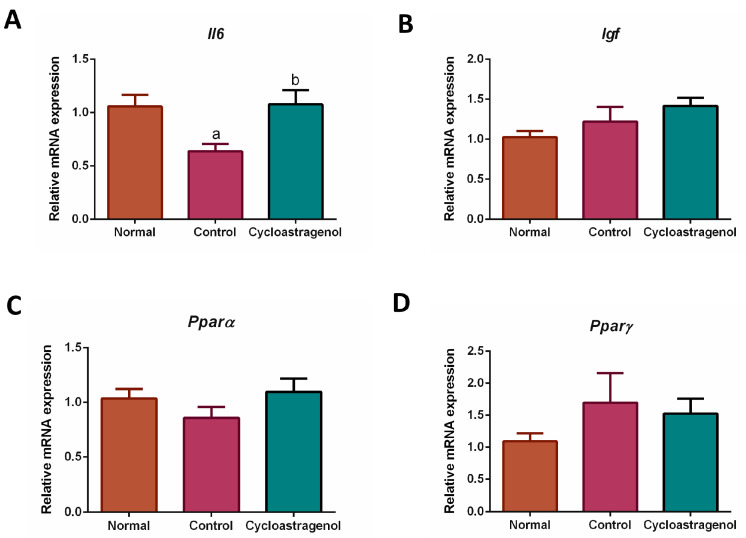
Effects of cycloastragenol (200 mg/kg) on the expression of gene-encoding markers associated with inflammation in the liver of mice treated with carbon-tetrachloride. Relative mRNA expression of interleukin 6 (*Il6*, (**A**)), insulin-like growth factor (*Igf*, (**B**)), peroxisome proliferator-activated receptor alpha (*Pparα*, (**C**)), and peroxisome proliferator-activated receptor gamma (*Pparγ*, (**D**)) when compared with the normal is shown. Bar graphs and corresponding error bars indicate means and SEM among the same treatment, respectively (*n* = 8–10). ^a^ and ^b^ indicate *p* < 0.05 when compared with the normal and control, respectively.

**Figure 7 biomedicines-11-00231-f007:**
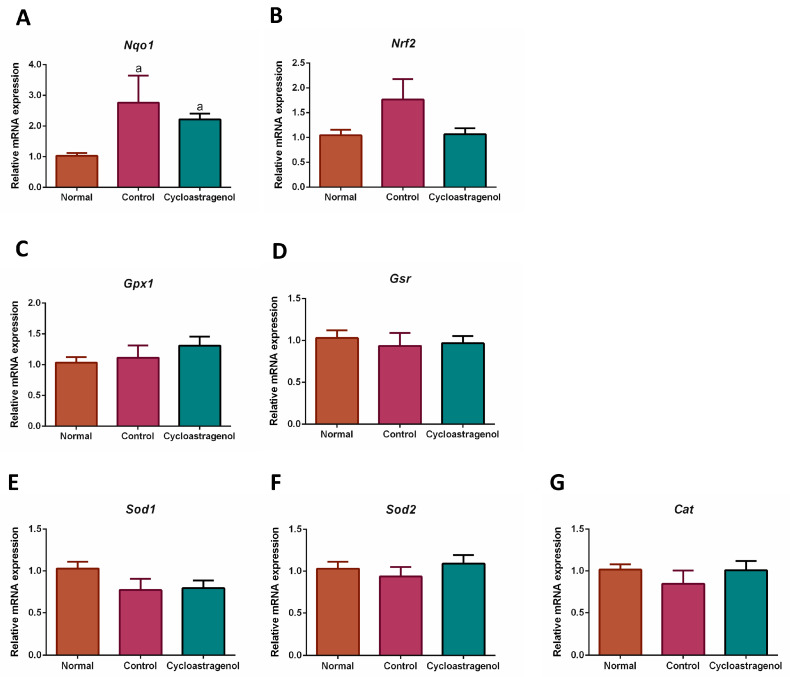
Effects of cycloastragenol (200 mg/kg) on the expression of gene-encoding markers associated with oxidative stress in the liver of mice treated with carbon-tetrachloride. Relative mRNA expression of NAD(P)H quinone dehydrogenase 1 (*Nqo1*, (**A**)), nuclear factor erythroid 2-related factor 2 (*Nrf2*, (**B**)), glutathione peroxidase 1 (*Gpx1*, (**C**)), glutathione-disulfide reductase (*Gsr*, (**D**)), superoxide dismutase 1 (*Sod1*, (**E**)), superoxide dismutase 2 (*Sod2*, (**F**)), and catalase (*Cat*, (**G**)) when compared with the normal is shown. Bar graphs and corresponding error bars indicate means and SEM among the same treatment, respectively (*n* = 8–10). ^a^ indicates *p* < 0.05 when compared with the normal.

**Figure 8 biomedicines-11-00231-f008:**
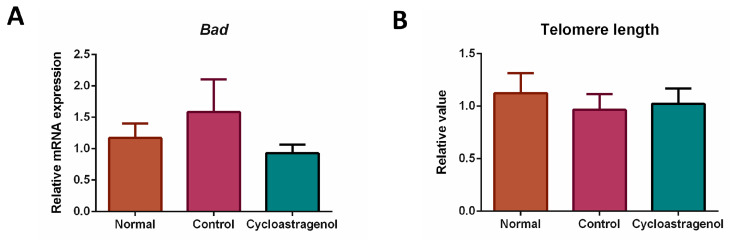
Effects of cycloastragenol (200 mg/kg) on the markers of senescence in the liver of mice treated with carbon-tetrachloride. Relative Bcl2-associated agonist of cell death (*Bad*) mRNA expression (**A**) and telomere length of genomic DNA (**B**) when compared with the normal are shown. Bar graphs and corresponding error bars indicate means and SEM among the same treatment, respectively (*n* = 8–10).

**Figure 9 biomedicines-11-00231-f009:**
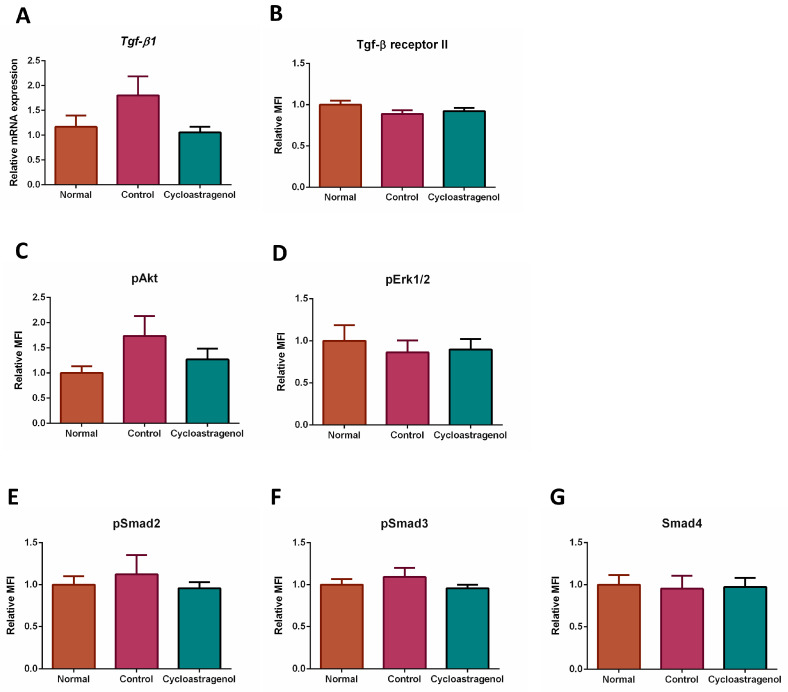
Effects of cycloastragenol (200 mg/kg) on the expression of markers associated with transforming growth factor-beta 1 (TGF-β1) signaling in the liver of mice treated with carbon-tetrachloride. Relative mRNA expression of TGF-β1 (*Tgf-β1*, (**A**)) and relative median fluorescence intensity (MFI) of transforming growth factor-beta receptor II (Tgf-β receptor II, (**B**)), phosphorylated protein kinase B (pAkt, (**C**)), phosphorylated extracellular signal-regulated kinase 1/2 (Erk1/2, (**D**)), phosphorylated-Smad2 (pSmad2, (**E**)), phosphorylated-Smad3 (pSmad3, (**F**)), and Smad4 (**G**) when compared with the normal are shown. Bar graphs and corresponding error bars indicate means and SEM among the same treatment, respectively (*n* = 8–10).

**Table 1 biomedicines-11-00231-t001:** Primer sequences used for quantitative real-time PCR.

Genes	Forward Primers (5′–3′)	Reverse Primers (5′–3′)
*Bad*	CTCCGAAGGATGAGCGATGAG	CTCCGAAGGATGAGCGATGAG
*Cat*	GGAGGCGGGAACCCAATAG	GGAGGCGGGAACCCAATAG
*Col1a1*	TGACTGGAAGAGCGGAGAGT	ATCCATCGGTCATGCTCTCT
*Gapdh*	ACAGTCCATGCCATCACTGC	GATCCACGACGGACACATTG
*Gpx1*	CCACCGTGTATGCCTTCTCC	AGAGAGACGCGACATTCTCAAT
*Gsr*	CACGGCTATGCAACATTCGC	GTGTGGAGCGGTAAACTTTTTC
*Igf*	TCGTGGGATGGGTGCTTT	TGAAGACAGTAGGGAAGAGACAAG
*Il6*	TCCATCCAGTTGCCTTCT	TAAGCCTCCGACTTGTGAA
*Nqo1*	AGGATGGGAGGTACTCGAATC	TGCTAGAGATGACTCGGAAGG
*Nrf2*	CTGAACTCCTGGACGGGACTA	CGGTGGGTCTCCGTAAATGG
*Ppar* *α*	CACTTGCTCACTACTGTCCTT	GATGCTGGTATCGGCTCAA
*Ppar* *γ*	GGTGCTCCAGAAGATGACAGA	TCAGCGGGTGGGACTTTC
*Sod1*	AACCAGTTGTGTTGTCAGGAC	CCACCATGTTTCTTAGAGTGAGG
*Sod2*	TGGACAAACCTGAGCCCTAAG	CCCAAAGTCACGCTTGATAGC
*Tgf-* *β* *1*	GGTTCATGTCATGGATGGTGC	TGACGTCACTGGAGTTGTACGG

## Data Availability

The data presented in this study are available on request from the corresponding author.

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
