# Peer review of "Hepatoprotective Efficacy of Cycloastragenol Alleviated the Progression of Liver Fibrosis in Carbon-Tetrachloride-Treated Mice"

_biomedicines, 2023, doi:10.3390/biomedicines11010231_

Round 1

Reviewer 1 Report

The manuscript is about the efficacy of cycloastragenol on carbon tetrachloride (CCl4)-induced liver fibrosis in mice. 

The manuscript is gut written.

Did the authors used olive oil vehicle in which CCL4 dissolved in the control group? and which vehicle is the cycloastragenol dissolved? Could you please mention the extraction method of cycloastragenol? 

It will be great if the size of the figures increased. 

Reviewer 2 Report

In this manuscript, the authors have investigated the impact of cycloastragenol as hepatoprotective agent against CCl4-induced liver fibrosis in mice animal model. Following my careful review, I can find that this study is well-conducted and organized, and the  used experimental groups are appropriate.  The experimental methods and analyzed parameters are suitable for this kind of studies. The results are appropriately described, and the figures are in a good order and well-prepared. The discussion is well written and explains the findings in a good order, and the conclusion is supported by the results. Therefore, I recommend this manuscript for publication following some minor corrections.

Generally, the abstract should be rewritten. It is not informative and does not reflect the aim of the research and the obtained results.  avoid use words such as our study, we found .....etc

In the introduction, some of the little flaws include, the lack of organization in presented information, while there are also many statements are without corresponding citation. ..............Lines 50-52 and other places in the whole manuscript.

Abbreviations should be fully described in the first mention

The authors must identify the type of anesthetic used and the technique of blood collection in the materials and methods section.

It is suggested to rename control group to be CCl4 group.

The authors should explain how they selected the dosages of cycloastragenol for this experiment.

The exact p value should be clear and mention in result section.

Number of animals should be added in legend to figures and it is suggested to add survival rate figure over time of the experiment and to know how many animal stay alive till the end of the experiment.  

It is mentioned in results section that there as an increase in liver enzymes however this not clear in the figures of AST and ALP no signs of any significance ??? please explain

In Figure 4 , histological findings presented for 3 groups not 4 group as in experimental design , please justify , the same for other figures one group is missing

 It would be helpful for the readers to discuss why cycloastragenol did not markedly regulate the expression of markers associated with canonical fibrogenic transforming growth factor-beta 1 signaling pathway
